# The violent death toll from the Iraq War: 2003–2023

Michael Spagat *

Department of Economics, Royal Holloway, University of London, Egham, United Kingdom

* m.spagat@rhul.ac.uk

## Abstract

From the beginning of the Iraq war, in March of 2003, to the present day, controversy has swirled around the death toll of the war. This paper narrows down the range of uncertainty for the numbers and trends in violent deaths in the war. I assemble and appraise all primary sources that cover the period from March of 2003 onwards—six sample surveys plus a casualty recording project (Iraq Body Count [IBC]). Data permitting, I present cumulative monthly figures with, for the surveys, 95% bootstrapped uncertainty intervals. The analysis uncovers a core of high-quality mainstream sources that are highly consistent with each another. In addition, there are three outlier surveys that are compromised by serious flaws and produce estimates far outside the mainstream. Discarding the outlying and flawed surveys reveals a clear picture of the violent death toll from the Iraq war. IBC figures, extended to include combatants, occupy a central position within the mainstream range of estimates. The strong consistency across the high-quality sources provides a rare validation of three war-death-measurement methodologies—household-based surveys, sibling-based surveys, and casualty recording. Methodological success notwithstanding, we must transcend the numbers to truly comprehend the human costs of the war.

## 1. Introduction

The Iraq War that began in March of 2003 has long been and remains a controversial subject (e.g., [1]). Indeed, the primary objective of [2–5] is to estimate the number of deaths in the war and these studies have received terrific attention with, respectively, 576, 865, 243 and 185 citations (57, 112, 41 and 80 since 2019) according to Google Scholar (checked 07/01/2024). Iraq Body Count (IBC) focuses strongly on violent deaths of civilians in the war and produces no fewer than 332,000 Google Scholar results (17,800 since 2019). In short, there exists quite a substantial, longstanding and continuing interest in the death toll from the Iraq War.

The attention lavished upon the above body of work has not, however, yielded a broad consensus on the quality of the sources of war deaths let alone on the number of people killed in the war. A recent survey on the collection of population health data both before and during the war [6] praises [2,3], cites [4] only in passing, ignores [5] and dismisses IBC. The Costs of War Project [7], in sharp contrast, notes that [2–4] were controversial and bases its account of the death toll during the first 20 years of the war on IBC. These clashing treatments of the primary

**Data Availability Statement:** All relevant data are within the manuscript and its Supporting Information files.

**Funding:** The author(s) received no specific funding for this work.

**Competing interests:** I am the Chair of the charity Every Casualty Counts which advocates for a practice, casualty recording, which is one of the methodologies evaluated in the paper." I now confirm that this declaration does not alter my adherence to PLOS ONE policies on sharing data and materials.

sources on deaths in the war are just two recent manifestations of a long-standing controversy that has sometimes been spirited (e.g., [8,9]).

I show in this paper that the primary sources on violent deaths in the Iraq war, including two further sources introduced below, split into a high-quality group that provides us with valuable evidence and a second incompatible group that should be dismissed. The existence of this latter group has generated considerable confusion with, for example, an article written for the 15th anniversary of the invasion characterizing the death toll as "still murky" [10]. This perceived murkiness may account for a noticeable tendency within the lessons-learned literature occasioned by the 20th anniversary of the invasion to formulate its lessons without reference to the death toll of the war (e.g., [11–13]).

The present paper clears up much of the confusion over the violent death toll from the Iraq war. A separate paper would be required to cover excess (including non-violent) deaths in the war although [14] already travels some distance in this direction.

The plan of the paper is as follows. The Materials and Methods section introduces the primary sources on violent deaths in the war, admitting all (and only) sources with coverage that begins from the onset of the war (March of 2003). Section 2 also explains the bases for the survey-based central estimates and 95% uncertainty intervals used in the paper. Sometimes data unavailability forces me to adopt published numbers rather than making my own calculations. However, whenever possible I produce my own central estimates with bootstrapped uncertainty intervals. Bootstrapping will yield accurate uncertainty intervals when the sample distribution mirrors the population distribution reasonably well, a condition that can fail in small samples, especially for highly skewed population distributions [15]. Therefore, for the [3] survey I provide a separate, wider, uncertainty interval derived in [15] that accounts for the skewed data collected in that small survey.

I also harmonize the estimates, to the extent possible, through two main measures. First, I use a single consistent series for the population of Iraq rather than the idiosyncratic population numbers used in each survey. Second, I extend IBC civilian-only figures to also include combatants, thus enabling me to compare like with like since none of the surveys distinguish between civilians and combatants and, therefore, they include both. The Methods and Materials section also evaluates the quality of the sources and even provides some new maps that demonstrate the poor quality of the sampling in [2]. I do raise some objections to upward adjustments made in the [4] survey however the truly serious quality problems I identify pertain just to [2,3] and the ORB survey (introduced below).

The Results section presents all the data, first in graphs and then in tables. These overviews effectively eliminate the "murkiness" referred to in [10]. They reveal, in particular, a clear mainstream of five highly consistent sources and an incompatible non-mainstream group of three outlier surveys that are precisely the ones that display serious weaknesses as explained in the Materials and Methods section. The truth on violent deaths in the Iraq War almost surely falls within the range spanned by the five mainstream sources that are displayed in Figs 4 and 5 below.

It is worth noting that @daponte2007 evaluated the available evidence way back in 2007, i.e., before two of the surveys covered in the present paper had been conducted and without the benefit of much of the negative evidence that eventually emerged against [2,3] and brought to bear in the present paper. Yet, without classifying the sources that did not exist in 2007, [16] already proposed the same grouping into reliable and unreliable sources that I advocate here.

The Discussion section summarizes and extends the main evidence and findings of the paper. By showing that three different methodologies for measuring violent war deaths have yielded measurements that are consistent with one another, the paper makes a key contribution to the field of war-death measurement. These methodologies are household-based surveys, sibling-based surveys and casualty recording as practiced by IBC (see [17–19]).

Importantly, the validation of household-based survey methodology only works to the extent that we can distinguish properly conducted surveys from improperly conducted ones. That is, the fact that we can find specific and serious flaws in the surveys that generate the non-main-stream estimates is good news for the war-deaths-measurement field as a whole because it locates the failure of these surveys in misapplications of survey methodology rather than in survey methodology itself.

An important contribution of the paper is to resolve the extraneous uncertainty that surrounds the violent death toll in the Iraq War stemming from the misinformation spread by the deficient surveys. This clarification constitutes a big step toward a proper understanding of the consequences of the momentous decision of the US to invade Iraq and is a crucial ingredient for drawing reliable lessons about the war.

The analysis shows that the IBC numbers, extended to included combatants, are a good guide to violent deaths through the middle of 2011, i.e., during the period for which surveys are available. Moreover, IBC methodology has remained stable so IBC is likely to continue being a good guide going forward after June of 2011. The Discussion section finishes with a back-of-an-envelope attempt to quantify uncertainty surrounding these post-June-2011 IBC-extended figures.

The Conclusion section sums the paper up and suggests going beyond the numbers in order to more deeply comprehend the true human costs of the war.

## 2. Materials and methods

This section introduces and evaluates the primary sources on post-2003 violent deaths in the Iraq war. These are [2–5,20–22]. I build violent-death figures for each source that are cumulative so as to maximize the opportunities for head-to-head comparisons across the sources. The appendix to the paper provides the R code that drives the analysis.

### Iraq Body Count (IBC) Extended

Here I will just sketch the IBC methodology. Readers with a strong interest should consult [19,23,24]. IBC documents violent deaths of civilians in the Iraq war using mainly media sources that are supplemented by, e.g., reports from NGO's, morgues, hospitals and the US military. Most database entries are dis-aggregated down to the incident level but many are composite figures that aggregate across multiple events, e.g., the number of violently killed individuals who passed through the Baghdad morgue during a full month (minus the number of specifically documented violent deaths recorded for Baghdad for the same month).

I began by downloading the publicly available IBC figures from the IBC website. These core IBC figures are for civilians only but the survey-based figures, described below, to which they are compared include combatants as well as civilians. I therefore bring the core IBC figures onto a comparable basis with the surveys by adding combatant figures to the downloaded civilian ones. I take most, but not all, of these combatant numbers from annual round-ups on the war that IBC posts on its website. However, I also plug a few gaps in these round-ups with other figures, most importantly those of [25] for the first ("shock and awe") phase of the war and those of [26] for 2017–2020 & 2022.

The end result of the data work, described in detail in the appendix, is monthly combatant plus civilian figures for the first 20 years of the war.

### Iraq Living Conditions Survey (ILCS)

The ILCS survey [21] recorded (among many other things) "war-related deaths" in 21,668 households spread across 2,200 clusters. I urge readers to consult the (copyrighted) map on

Page 12 of [27]. It shows the cluster locations for the ILCS matching up very well with a night-lights map of Iraq, thus demonstrating the excellent representativeness of the sample.

The ILCS published an estimate of 24,000 war-related deaths, with a 95% uncertainty interval ranging from 18,000 to 29,000, that covers a period from the beginning of the war in March of 2003 through May of 2004. There is, unfortunately, no publicly available ILCS data set for me to work with so I use these published figures.

## Roberts et al. 2004 [2]

[2] recorded violent (and non-violent) deaths in 983 households spread across 33 clusters. Fig 1 displays cluster locations for the survey together with primary roads for Iraq. Publicly released data, provided in the appendix of the present paper, specifies the locations of two triples of clusters (in the governorates of Babylon and Mayson) only at the governorate level. So in Fig 1 I assigned random locations to these clusters within each governorate.

The tendency to sample close to a few primary roads is accentuated if we locate the Babylon and Mayson clusters in, respectively, the governorate capitals of Hillah and Amarah as Fig 2 does.

The fact that many clusters with well-specified locations are placed within governorate capitals lends credibility to Fig 2 as a plausible guide to all the cluster locations in the [2] survey. However, even if Fig 1 is closer to the truth than Fig 2 is it is still clear that one field team followed a northern loop running from Baghdad to Balad to Baiji to Shirqat to Mosul to Sulaimaniya to Muqdadiya to Baqubah and back to Baghdad (or vice versa) while a southern team went from Baghdad to Karbala to Hillah (with possible detours) to Rifa to Shatra to Suq Al-Shoyokh to Amarah(with possible detours) to Kut to Baghdad (or vice versa). Another team made a quick round trip to Fallujah and back to Baghdad without penetrating any deeper into Anbar governorate.

The above mapping exercise demonstrates that the 33 cluster locations of [2] are not representative of the country as a whole. Indeed, there are no clusters whatsoever in the governorates of Basrah, Qadisiyah, Najaf, Salah Al-Din, Arbil, Duhok or Al-Muthanna, which are shown in green in Fig 3:

This convenience sampling along a handful of primary roads is consistent with the fact that all interviews are reported to have been completed in a very short span of time between September 8 and 20 ([2], p. 1860). [28] argued that the survey of [3] (covered below) was upward biased because the final stage of its sampling scheme started with main streets which are likely to be unrepresentatively violent ("main-street bias"). A version of this critique applies with particular force at a macro level to [2] given that all of its clusters seem to have been located along the paths of a few primary roads. In short, [2] is afflicted by "primary road", rather than "main-street", bias. I judge this sampling bias to be severe but simply note the problem without attempting to adjust for it.

There is sufficient available data for [2] to allow me to make my own estimate of 260,000 violent deaths with a 95% (bootstrapped) uncertainty interval running from 65,000 to 640,000. This gargantuan range suggests that the primary-road-bias critique may be beside the point. The extreme factor-of-10 uncertainty surrounding the above estimate renders it as a virtual non-measurement, akin to a prediction that tomorrow's temperature will be between -20 and 50 degrees centigrade.

## Burnham et al. (2006) [3]

Gilbert Burnham, the principal researcher on the [3] survey was formally censured in 2009 by the American Association for Public Opinion Research (AAPOR) for refusing to disclose

## Primary Roads and the Sampling Locations for Roberts et. al (2004)
Only locations specified below the governorate level are labelled

Note - The two triples of unlablled points are random locations in Babylon (left) and Mayson (right) governorates

**Fig 1. The [2] survey stuck close to a few primary roads.**

essential facts about the survey's methodology. Richard Kulka, the president of AAPOR at the time, issued the following summary statement to accompany the censure:

> "When researchers draw important conclusions and make public statements and arguments based on survey research data, and then subsequently refuse to answer even basic questions about how their research was conducted, this violates the fundamental standards of science, seriously undermines open public debate on critical issues, and undermines the

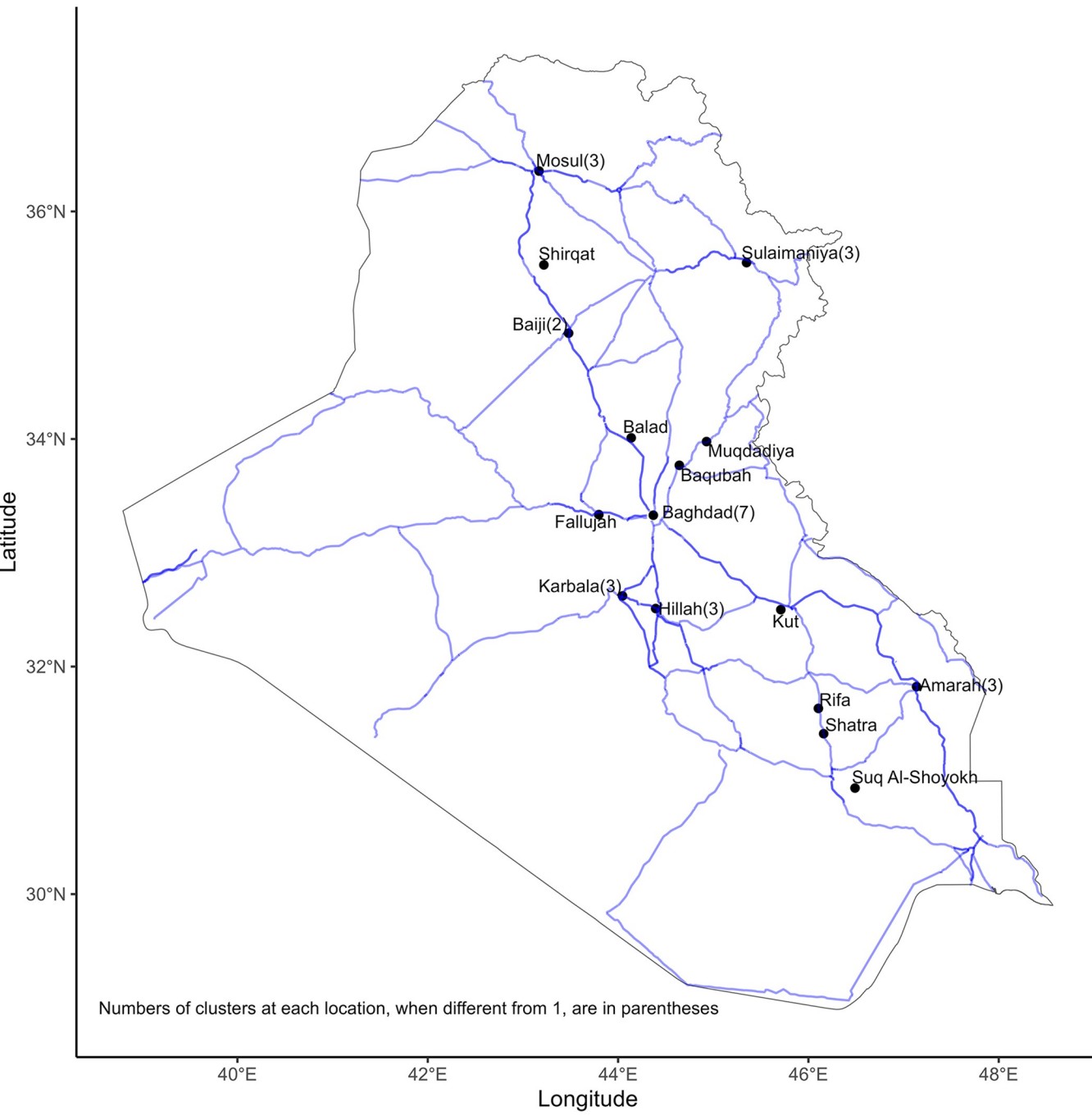

## Primary Roads and the Sampling Locations for Roberts et. al (2004)
Sampling Locations in Babylon and Maysan are Placed in the Seat of each Governorate

Numbers of clusters at each location, when different from 1, are in parentheses

**Fig 2. Cluster locations with Babylon and Mayson clusters placed in Hillah and Amarah, respectively.**

credibility of all survey and public opinion research. These concerns have been at the foundation of AAPOR's standards and professional code throughout our history, and when these principles have clearly been violated, making the public aware of these violations is an integral part of our mission and values as a professional organization." [29]

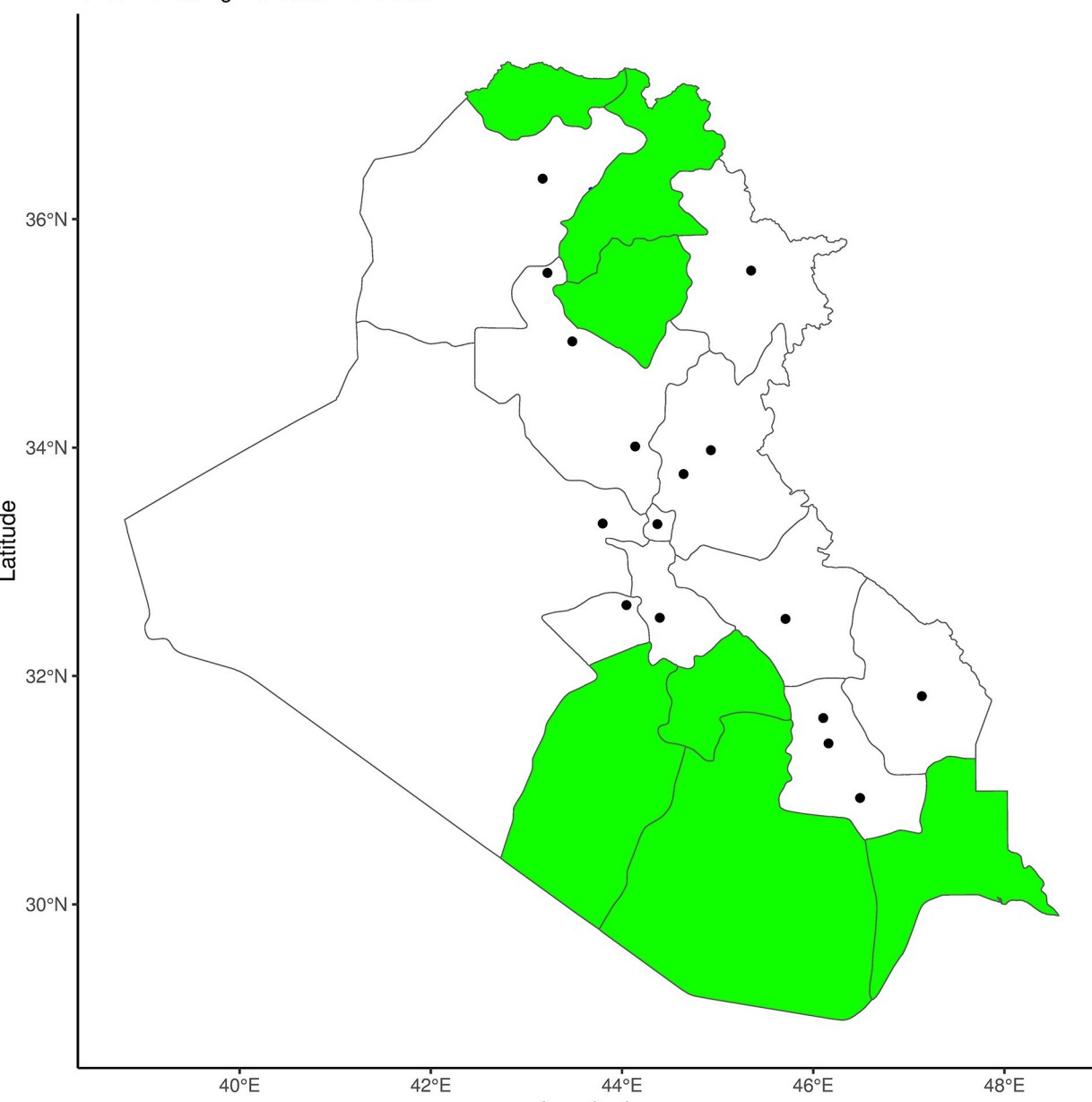

**Fig 3. There are no clusters in large contiguous areas in both the North and the South.**

Moreover, [30] presents evidence of data fabrication and ethical violations in the [3] survey. Burnham's university, Johns Hopkins, did not investigate the data fabrication evidence but did launch an investigation into the ethics of the project which led to the suspension of Gilbert Burnham because he collected unique identifiers for his interviewees despite obtaining light-touch review from his IRB by promising not to collect unique identifiers [31]. [16,32] also found serious shortcomings in the survey.

Despite these shadows that hang over the [3] survey, I include it in the present paper so that readers can place its ostensible findings within the context of all the primary evidence on violent deaths in the Iraq War. This bird-eye view reveals the violent-death estimates based on this survey to be extreme outliers.

[3] reportedly interviewed 1,849 households spread over 47 clusters about violent (and non-violent) deaths from the beginning of the war through June of 2006. One cluster, nonetheless, lists no fewer than 24 violent deaths in July of 2006, a month outside the coverage period for the survey. [3] includes these out-of-range deaths in the paper's estimates and I follow this lead for my estimates. However, some readers may wish to subtract about 50,000 July-2006 deaths from my estimates to compensate for this coverage-period anomaly.

I am able to make monthly central estimates based on the data of [3], a level a detail that is not possible for the [21] or the [2] surveys. However, I can only construct an uncertainty interval around the central estimate for the whole time period covered by the survey because this is the only period for which I have cluster-by-cluster data. My central estimate for the whole period is 670,000 with a 95% uncertainty interval running from 440,000 to 940,000.

[15] point that [3] have a small sample for which the distribution of violent deaths by cluster is highly skewed. They argue that bootstrapped uncertainty intervals are, therefore, likely to be too narrow. They propose a special methodology for calculating uncertainty intervals in such cases which, when applied to the [3] data results in a 95% uncertainty interval running from 293,339 to 988,101 (rounded to the nearest 10,000). We will use both this interval and the bootstrapped on below.

## Iraq Family Health Survey (IFHS)

The IFHS interviewed 9,345 households spread over 971 clusters and published a violent-death estimate covering the beginning of the war through June of 2006 in [4]: 151,000 with a 95% uncertainty interval running from 104,000 to 223,000.

The time period covered by the IFHS is identical to that of [3], aside from the latter's July 2006 anomaly. Yet, the bottom of the bootstrapped 95% uncertainty interval for [3] is nearly twice the top of the 95% uncertainty interval for [4] and even the bottom of the [15] uncertainty interval does not overlap with the IFHS one. So at least one of the two surveys must badly mis-measure the number of violent deaths in the Iraq War for this common period. The obvious candidate to be wrong is [3], given the evidence of fabrication, non-transparency over its methods and ethical violations that suggest lack of control over the field work.

Yet there are two reasons to believe that even the [4] survey overestimated violent deaths; correcting these problems would pull it even further below the [3] estimates. First, [4] uses the following weak argument to multiply what would have been a conventional survey-based violent-death estimate by more than a factor of 1.6:

> "In general, the underreporting of deaths is likely to be common in household surveys. The most serious concern is household dissolution after the death of a household member. Several demographic assessments have suggested that there has been an underreporting of deaths in the IFHS. The application of the growth balance method,[7] with the use of the age distribution of deaths in the population obtained from the household roster, indicates that the level of completeness in the reporting of death was 62%. However, this estimation needs to be interpreted with caution, since a basic assumption of the method—a stable population —is violated in Iraq. Furthermore, the comparison is not made to a rate of death derived from two successive censuses, as is usually done, but from the age distribution of the households in the IFHS."

Moreover, this large upwards adjustment, aptly labelled an "arbitrary fudge" [33], is applied on top of an earlier adjustment to account for clusters that were selected for interviews but were not ultimately included in the sample because of security concerns that arose during the field work. The motivation for the missing-cluster adjustment is much stronger than the motivation for the factor-of-1.6 adjustment. However, the specific adjustment that is applied assumes that missing clusters in Baghdad were roughly four times as violent as the completed clusters throughout the whole course of the war. This big adjustment is justified on the ground that it brings the IFHS ratio of Baghdad deaths to deaths in a basket of relatively peaceful governorates into rough equality with the same ratio for IBC. However, this adjustment will be excessive to the extent that IBC is relatively better at recording Baghdad deaths than it is at recording deaths outside of Baghdad.

The two upward adjustments combined roughly double the violent-death estimate of [4]. It is probably going too far to totally eliminate both of them but, at a minimum, awareness of their existence should make us cautious about the adjusted IFHS estimate. In the tables and graphs below I include both adjusted and unadjusted IFHS figures although I cannot produce uncertainty intervals for the unadjusted ones. As always, the appendix provides details including R code.

## Opinion Research Business (ORB) Survey

The internet no longer contains primary material on the ORB survey so the best way to learn about it is through a critique, reply and rejoinder published in Survey Research Methods [34,35].

ORB initially announced in October 2007 that they had interviewed 1,499 people spread across an undisclosed number of clusters and made an estimate of 1,229,580 violent deaths with a range of 733,158 to 1,446,063 covering a period from the beginning of the war through August 2007. In January 2008 ORB announced that it had done further rural sampling and that:

> "... we now estimate that the death toll between March 2003 and August 2007 is likely to have been of the order of 1,033,000. If one takes into account the margin of error associated with survey data of this nature then the estimated range is between 946,000 and 1,120,000." ORB (2008a)

ORB added that this estimate was based on 2,163 interviews spread over 112 "sampling points", i.e., clusters. Note that the published range cannot be a proper 95% uncertainty interval, given the small number of clusters in the sample.

There are remarkable anomalies in the ORB survey which emerge when it is placed between two other ORB surveys conducted in Iraq before and after the one that yielded the estimate of 1 million violent deaths [34]. The before and after surveys supposedly ask about deaths of *family* members whereas the middle survey (responsible for the 1 million estimate) supposedly asks about deaths within *households*. Household is a much narrower category than family, the latter of which can include brothers, sisters, aunts, uncles, cousins, grandparents, etc., all people who might not live within the immediate household. Sensibly, in southern governorates the fraction of interviewees that reported violent deaths decreased by a factor of 5 in the middle survey compared to the first one. However, in four central governorates that account for about 80% of the estimated 1 million deaths the fraction of respondents reporting deaths in the middle, supposedly household-based survey, is substantially *higher* than the fraction reporting deaths in the first, supposedly family-based, survey. The third survey delivers a further blow to

the credibility of ORB's work in Iraq; interviewers supposedly switch to asking about families again but the fractions reporting deaths are roughly equal in the South for the middle and third survey.

In short, ORB's Iraq work is not credible. Nevertheless, I still include it in the present paper, again to place it into perspective. I use ORB's published figures since ORB has not released the data that I would need to make an independent estimate.

### University Collaborative Iraq Mortality Study (UCIMS)

The UCIMS, written up in [5], is by far the best of the surveys according to the criterion of data openness. I am able to make monthly UCIMS-based violent-death estimates complete with 95% bootstrapped uncertainty intervals. Moreover, I can make a second set of estimates based on the sub-category of reported deaths that were backed by death certificates. Still better, I can make separate monthly violent-death estimates with 95% bootstrapped uncertainty intervals based on a different module in the survey's questionnaire that asks about deaths of siblings rather than deaths of household members.

The UCIMS interviewed 1,976 households spread across 100 clusters about violent (and non-violent) deaths from the beginning of the war through June of 2011. My central estimates for violent deaths for this whole period are 200,000, 140,000 and 140,000 based on, respectively, all reported violent deaths within households, only within-household deaths backed by death certificates and reported violent deaths of siblings. The 95% uncertainty intervals for the three estimates are, respectively, 140,000 to 270,000, 100,000 to 180,000 and 110,000 to 180,000. I give the above estimates for orientation but the graphs of the next section show cumulative estimates with uncertainty intervals for every month covered by the survey. Note that [5] did not publish a proper violent-death estimates based on the household module of their survey but they did publish a central estimate of 132,000 with a 95% uncertainty interval of 89,000 to 174,000 based on the sibling module, numbers that are fairly close to my sibling-based estimates. As always, the details are in the appendix.

## 3. Results

### Graphical comparisons

Fig 4 shows cumulative figures covering the first 20 years of the war for IBC, ILCS, [2,3], IHFS-adjusted, ORB, UCIMS-household (accepting all reported deaths regardless of death-certificate backing) and UCIMS-sibling. I show all data at the lowest possible level of aggregation, down to the month level if possible. The picture also displays 95% uncertainty intervals. For [3] I show both my bootstrapped interval (the narrower one) and the [15] interval (the wider one). I use shading for the two UCIMS-based estimates and line segments otherwise.

The violent-death estimates for the Iraq war divide into two groups. Group 1 consists of the ILCS, IFHS-adjusted, UCIMS (household and sibling) and IBC-extended. None of these sources are above criticism, of course, and I did suggest that at least part of the upward adjustment made to the IFHS in [4] went too far. However, none of them have glaring deficiencies comparable to those of the group-2 surveys: [2,3] and the ORB survey. The latter the three surveys are the ones identified as particularly problematic in the previous section: [2] for sampling just along a few primary roads and a terrifically wide uncertainty interval, [3] for non-transparency, data fabrication and ethical violations and ORB for discrediting irregularities across multiple surveys. Recall, moreover, that the ILCS has by far the biggest sample of all the surveys and the figure on page 12 of [27] demonstrates that this sample represents the population of Iraq well.

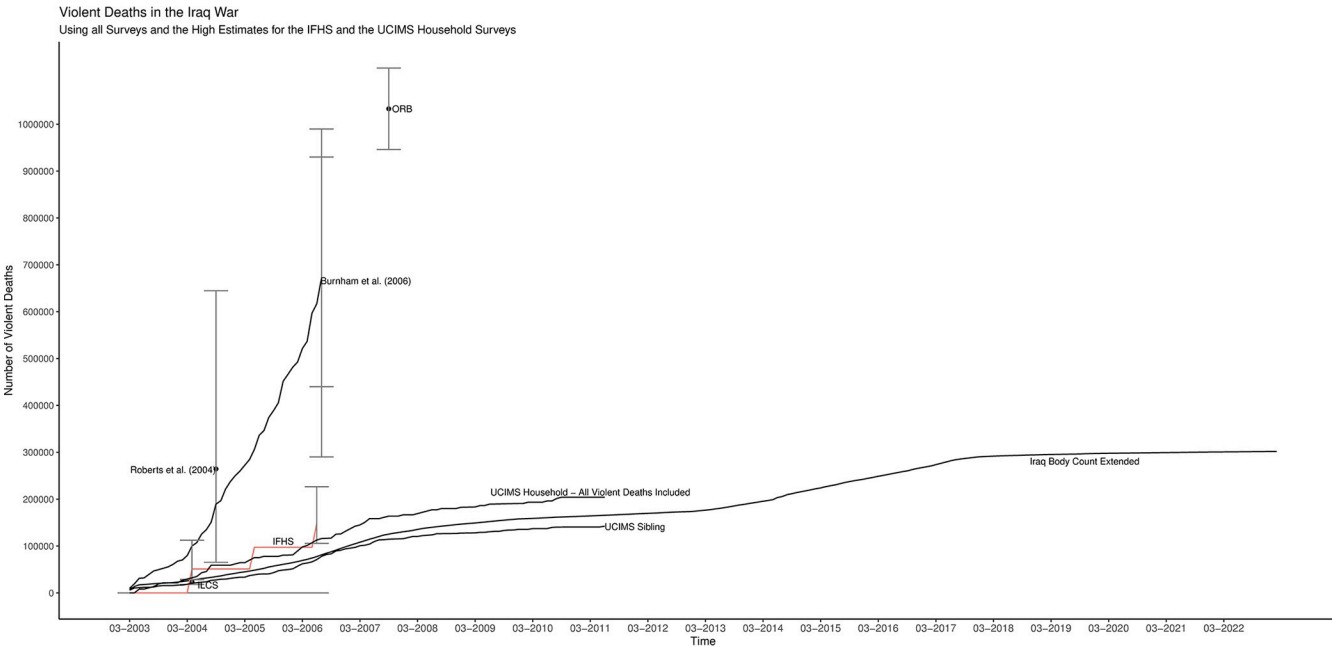

**Fig 4. Violent-death numbers for the Iraq war divide into two groups.**

Fig 5 shows just the group-1 sources and uses the unadjusted IFHS figures along with the version of the UCIMS-household figures that require death-certificate verification for each death. Again, all figures are cumulative and I show 95% uncertainty intervals where possible, using either line segments or shading (for the UCIMS). These five sources, different in two cases from the group-1 sources in Fig 4, continue to tell a consistent story.

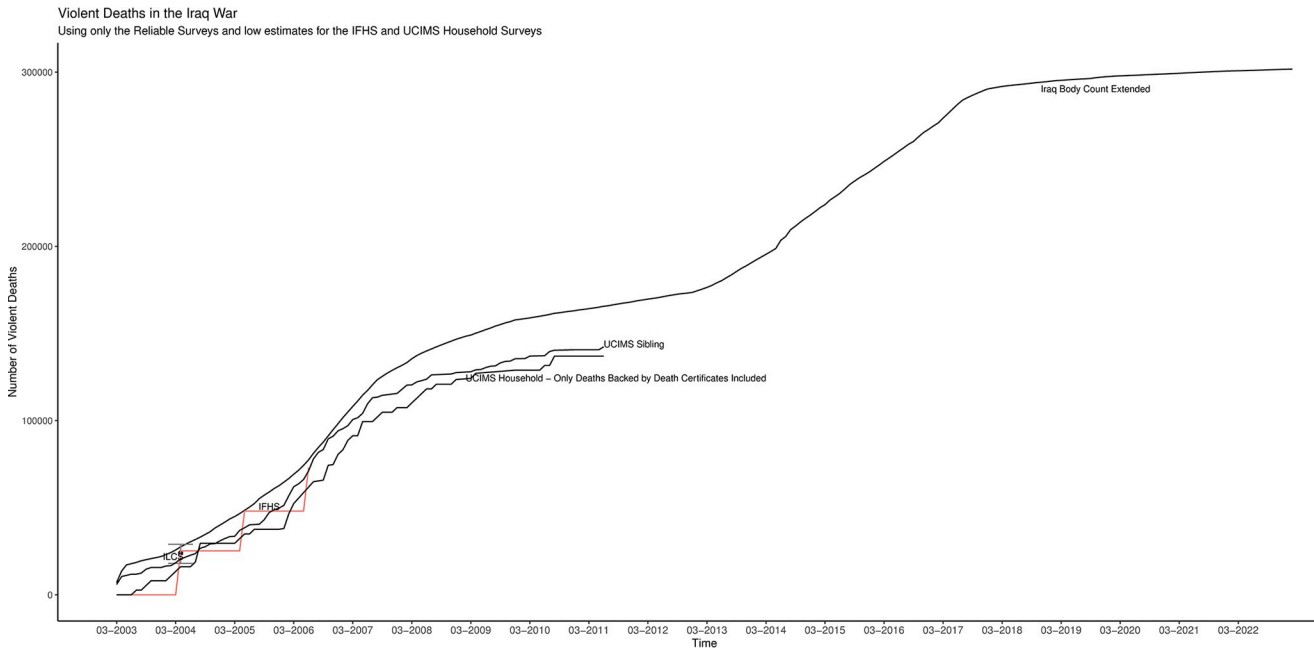

**Fig 5. A consistent picture of cumulative violent deaths in the Iraq War.**

**Table 1. Violent deaths in the Iraq War March 2003 through April 2004.**

| Sources | Percentile 2.5 | Central Number | Percentile 97.5 | Percentile 100 |
|---|---|---|---|---|
| Mainstream | | | | |
| UCIMS Household—Deaths Backed by Death Certificates | 5,400 | 16,000 | 30,000 | 38,000 |
| UCIMS Sibling | 14,000 | 21,000 | 28,000 | 34,000 |
| ILCS | 18,000 | 24,000 | 29,000 | NA |
| IFHS Unadjusted | NA | 25,000 | NA | NA |
| IBC Extended | NA | 28,000 | NA | NA |
| UCIMS Household—All Deaths | 13,000 | 32,000 | 54,000 | 75,000 |
| On the Edge | | | | |
| IFHS Adjusted | 28,000 | 51,000 | 110,000 | NA |
| Outside the Mainstream | | | | |
| Burnham et al. (2006) [3] | NA | 100,000 | NA | NA |

## Tabular comparisons

I turn to tables in the present sub-section for two main reasons. First, the pictures in the previous sub-section are challenging to read in areas where there are multiple estimates and uncertainty intervals in close proximity to each other. Second, the tables contain information on the bootstrap runs that is not present in the pictures in the previous sub-section.

All tables present cumulative figures that cover periods from the beginning of the war in 2003 and stop at the following endpoints: April 2004, September 2004, May 2005, June/July 2006 and August 2007. These endpoints are dictated by the periods covered by the various surveys.

Table 1 gives central estimates and 95% bootstrapped uncertainty intervals, when possible, together with the IBC-extended number for violent deaths in the Iraq War from March 2003 through April 2004. It also provides the maximum estimate over 10,000 bootstraps whenever bootstrapping is possible; this is new information that was not in the graphs.

There is a mainstream of six sources with central estimates ranging between 16,000 and 32,000 violent deaths and no instances of the bottom of a 95% uncertainty interval coming remotely close to exceeding the top of another interval within this mainstream. I classify the IFHS adjusted estimate as "on the edge" because the bottom of its 95% uncertainty interval is close to exceeding the tops of three of the intervals that are within the mainstream. I classify [3] as outside the mainstream because its central estimate for the period exceeds by wide margins even the maximum bootstrap outcomes in all cases for which bootstrapping is possible.

Table 2 covers the period from the beginning of the war through September of 2004. The central estimate for [3] is, again, well above the highest of 10,000 bootstrap estimates whenever

**Table 2. Violent deaths in the Iraq War March 2003 through September 2004.**

| Sources | Percentile 2.5 | Central Number | Percentile 97.5 | Percentile 100 |
|---|---|---|---|---|
| Mainstream | | | | |
| UCIMS Sibling | 19,000 | 28,000 | 36,000 | 43,000 |
| UCIMS Household—Deaths Backed by Death Certificates | 13,000 | 30,000 | 48,000 | 70,000 |
| IBC Extended | NA | 35,000 | NA | NA |
| UCIMS Household—All Deaths | 30,000 | 59,000 | 94,000 | 130,000 |
| Outside the Mainstream | | | | |
| Burnham et al. (2006) [3] | NA | 190,000 | NA | NA |
| Roberts et al. (2004) [2] | 65,000 | 260,000 | 640,000 | 1,200,000 |

**Table 3. Violent deaths in the Iraq War March 2003 through May 2005.**

| Sources | Percentile 2.5 | Central Number | Percentile 97.5 | Percentile 100 |
|---|---|---|---|---|
| Mainstream | | | | |
| UCIMS Household—Deaths Backed by Death Certificates | 16,000 | 35,000 | 59,000 | 83,000 |
| UCIMS Sibling | 26,000 | 38,000 | 52,000 | 67,000 |
| IFHS Unadjusted | NA | 48,000 | NA | NA |
| IBC Extended | NA | 48,000 | NA | NA |
| UCIMS Household—All Deaths | 40,000 | 75,000 | 120,000 | 150,000 |
| IFHS Adjusted | NA | 97,000 | NA | NA |
| Outside the Mainstream | | | | |
| Burnham et al. (2006) [3] | NA | 310,000 | NA | NA |

bootstrapping is possible. The central estimate for [2] is very much higher than even the [3] estimate although the 95% uncertainty interval for the former estimate is so wide that it almost reaches down as far as the central estimate for the UCIMS household survey when all reported deaths, regardless of death-certificate confirmation, are included.

Table 3 presents all the available evidence covering the period from the beginning of the war through May of 2005. This time I have placed the adjusted IFHS estimate within the mainstream since its central estimate is substantially closer to the mainstream than was the case in Table 1. [3], on the other hand, is even further outside the mainstream than it was in Table 1 with a central estimate more than double the largest bootstrap estimate obtained within the mainstream.

Table 4 covers the period from the beginning of the war through the middle of 2006. Now I am able to do bootstrapping for [3], revealing that even the bottom of its 95% uncertainty is more than double the largest bootstrap simulation within the mainstream. Even the bottom of the exceptionally wide uncertainty interval of [15] still exceeds the top of the IFHS interval by 60,000 deaths. In fact, the minimum bootstrap estimate for [3] is 270,000, a number that exceeds the maximum bootstrap estimate for the mainstream sources by 70,000 deaths. In short, there is radical incompatibility between [3] and the mainstream.

Finally, Table 5 displays the available information for the period from the beginning of the war through August of 2007.

To summarize, the tables support the idea that there is a mainstream of estimates that extends from the household-based UCIMS estimates that count only deaths backed by death certificates up to the adjusted IFHS estimates. These are the sources designated as group 1 in the previous section.

**Table 4. Violent deaths in the Iraq War March 2003 through June/July 2006.**

| Sources | Percentile 2.5 | Central Number | Percentile 97.5 | Percentile 100 |
|---|---|---|---|---|
| Mainstream | | | | |
| UCIMS Household—Deaths Backed by Death Certificates | 35,000 | 62,000 | 94,000 | 130,000 |
| UCIMS Sibling | 50,000 | 71,000 | 95,000 | 110,000 |
| IFHS Unadjusted | NA | 73,000 | NA | NA |
| IBC Extended | NA | 77,000 | NA | NA |
| UCIMS Household—All Deaths | 67,000 | 110,000 | 160,000 | 200,000 |
| IFHS Adjusted | 110,000 | 150,000 | 230,000 | NA |
| Outside the Mainstream | | | | |
| Burnham et al. (2006) [3] | 440,000 | 670,000 | 930,000 | 1,300,000 |
| Burnham et al. data—Rosenblum and van der Laan UI | 290,000 | NA | 990,000 | NA |

**Table 5. Violent deaths in the Iraq War March 2003 through August 2007.**

| Sources | Percentile 2.5 | Central Number | Percentile 97.5 | Percentile 100 |
|---|---|---|---|---|
| Mainstream | | | | |
| UCIMS Household—Deaths Backed by Death Certificates | 67,000 | 100,000 | 150,000 | 170,000 |
| UCIMS Sibling | 85,000 | 110,000 | 150,000 | 180,000 |
| IBC Extended | NA | 130,000 | NA | NA |
| UCIMS Household—All Deaths | 110,000 | 160,000 | 220,000 | 280,000 |
| Outside the Mainstream | | | | |
| ORB | 950,000 | 1,000,000 | 1,100,000 | NA |

The ORB survey is far out of the mainstream.

For periods of the Iraq war covered by one of the above tables it is likely that the true number of violent deaths lies within or near to the mainstream figures shown in the tables. It is bad practice to cite just a single figure for any period but, if forced, I would choose the IBC-extended number which is always at the heart of the mainstream. After June of 2011 we are left with only the IBC-extended figures. Based on the consistent progression of IBC-extended through the middle of the group-1 surveys during the periods covered by the surveys and the fact that IBC did not change its methodology after 2011 I would suggest that IBC-extended remains a good guide to the violent death toll of the war after June of 2011.

## 4. Discussion

The present paper provides a rare validation for three different methodologies: household-based survey estimation [4,5,21], sibling-based survey estimate [5] and casualty recording (IBC). The only comparable validation I am aware of is [36] which finds strong consistency between three accounts of violent deaths in the Kosovo war that use three different methodologies. These methodologies included a household survey and a casualty recording project so this Kosovo validation enhances confidence in the results of the present paper. More such validations in the future would enhance confidence both in the results of the present paper and in the methods used for measuring violent war deaths.

Discussions of the death toll in the Iraq war have long been complicated and confused by the presence of a parallel universe of outlying estimates (group 2) that are inconsistent with the mainstream universe (group 1). But this confusion evaporates once we discard group 2, a move that is supported both by quality assessments of the sources and the weirdness of the numbers. Indeed, if there were not serious and identifiable shortcomings in the group-2 surveys we would have to acknowledge the methodology of survey-based measurement of war deaths to be unreliable to the point of unusability. Similarly, we would have to discard a body-temperature-measuring device if, when used as intended, can measure 100 degrees one moment and 90 degrees the next. Thus, the credibility of the survey methodology for measuring violent war deaths hinges on our ability to separate improper applications of the methodology from proper ones. Fortunately, it turns out that we are able to do this.

The present paper clears up extraneous uncertainty over violent deaths in the Iraq war between March of 2003 and June of 2011 when we have not only the IBC-extended numbers but also good surveys using multiple methods and treatments, e.g, household vs. sibling, adjusted versus unadjusted and many time periods. Nevertheless, considerable uncertainty remains which we have quantified through 95% uncertainty intervals for individual surveys when these exist. Yet each such uncertainty interval is calculated in isolation from the other

group-1 evidence, thereby ignoring the great consistency across these sources and probably exaggerating the remaining uncertainty.

IBC is the sole remaining source covering the whole country after June of 2011. We cannot, therefore, cross validate the IBC-extended numbers for the post-June-2011 period as we did for the pre-June-2011 period. Nevertheless, the analysis of the paper consistently locates IBC-extended squarely within the range of the mainstream sources when group-1 surveys exist and IBC methodology did not change in July of 2011. So it is reasonable to accept the IBC-extended numbers as a good guide to the post-June-2011 violent death toll in the war.

The absence of post-June-2011 surveys imposes the further cost of denying us an obvious basis for quantifying the uncertainty surrounding the IBC-extended figures as these are not sample-based estimates. I cautiously fill this gap with a back-of-the-envelope calculation based on the following observation. During the pre-June-2011 period, i.e., when there are surveys, we have 95% confidence intervals surrounding these surveys. The IBC-extended figures go right through the middle of the central estimates of these surveys so we can take the uncertainty intervals surrounding the survey central estimates as characterizing the uncertainty around the IBC figures up until June-2011. I then project this uncertainty forward. The ratios of IBC-extended figures to the central estimates for each treatment of the group-1 surveys for the whole time period covered by each survey are 0.83, 0.86, 0.87, 0.94, 1.2 and 1.9 for, respectively, UCIMS-household requiring death certificate backing, UCIMS-sibling, ILCS, IFHS-unadjusted, UCIMS-household including all violent deaths and IFHS-adjusted. These numbers lead to the following tentative conclusions about the human cost of the first 20 years of the war; 1. The IBC-extended figures, roughly 200,000 violent deaths might be 20% higher than the true numbers of violent deaths: 2. The true numbers of violent deaths might, at a stretch, be twice the IBC-extended Fig 3. The mean of the above ratios is 1.1 while the median ratio is 0.91 so the preponderance of evidence points toward the IBC-extended figures being about right.

Although I have drawn some fairly strong conclusions from the analysis it is important to review the main limitations of this study which divide broadly into issues with the quantity and the quality of the data. On the quantity issue, I note that more data, especially after 2011, would be welcome as we have only Iraq Body Count after that. But more pre 2011 data would also be welcome.

The main quality issue concerns the extreme unreliability of the (Roberts et al. 2004), (Burnham et al. 2006) and ORB estimates. Much effort in the paper is required to dispel the misinformation created by these works. However, quality issues are not confined to these surveys. For example, all field teams worked under time pressure because they were operating in time of war and the Iraq Family Health Survey even had to abandon some of its clusters due to security problems. A further issue is that several of the surveys have not made their data available.

## 5. Conclusion

The above analysis validates the choice of the Costs of War Project [7] to use IBC as the basis for its quantification of the human cost of the Iraq War and to give short shrift to the group-2 surveys favored by [6]. Such quantification projects must work with the best available evidence and should not squander their credibility through exaggeration as, e.g., [37] do with their estimate of 2.4 million deaths in the Iraq war with a range of 1.5 million to 3.4 million.

Inevitably, some scientific papers are wrong and public trust in science requires that these errors are exposed and discarded. The existence of dueling survey-based estimates of violent deaths in the Iraq war has damaged the credibility of the field while the present paper should

help to restore some of this precious commodity. Transparency is a central ingredient to the scientific vetting process, and I have relied on the availability of some, alas not all, of the crucial data in writing the present paper. Hopefully, the field of war-death estimation will move toward greater transparency in the future so that more work along the lines of the present paper will be possible.

[38,39] argue that good quantification is important yet inadequate to the task of illuminating the horror of warfare. They advocate telling the stories of the many individual victims who lie beneath the abstract numbers. [19] does precisely this, telling the stories of many of the individuals whose deaths are incorporated into the IBC database. The present paper's validation of IBC data can only enhance the value of this work because it shows that [19] provides, broadly, an overview of the stories of all victims in the war.

## Supporting information

**S1 Data.**
(ZIP)

## Acknowledgments

I would like to thank Michael Robbins for useful feedback on this paper.

## Author Contributions

**Conceptualization:** Michael Spagat.

**Data curation:** Michael Spagat.

**Formal analysis:** Michael Spagat.

**Investigation:** Michael Spagat.

**Methodology:** Michael Spagat.

**Project administration:** Michael Spagat.

**Resources:** Michael Spagat.

**Validation:** Michael Spagat.

**Visualization:** Michael Spagat.

**Writing – original draft:** Michael Spagat.

**Writing – review & editing:** Michael Spagat.

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
