## [Decision Letter · Decision Letter 0]

20 Sep 2023

PONE-D-23-23835The Violent Death Toll from the Iraq War: 2003 - 2023PLOS ONE

Dear Dr. Spagat,

Thank you for submitting your manuscript to PLOS ONE. After careful consideration, we feel that it has merit but does not fully meet PLOS ONE’s publication criteria as it currently stands. Therefore, we invite you to submit a revised version of the manuscript that addresses the points raised during the review process.

We look forward to receiving your revised manuscript.

Kind regards,

Masoud Behzadifar

Academic Editor

PLOS ONE

Journal Requirements:

   "I am the Chair of the charity Every Casualty Counts which advocates for a practice, casualty recording, which is one of the methodologies evaluated in the paper."

3. We note that Figures 1-3 in your submission contain [map/satellite] images which may be copyrighted. All PLOS content is published under the Creative Commons Attribution License (CC BY 4.0), which means that the manuscript, images, and Supporting Information files will be freely available online, and any third party is permitted to access, download, copy, distribute, and use these materials in any way, even commercially, with proper attribution. For these reasons, we cannot publish previously copyrighted maps or satellite images created using proprietary data, such as Google software (Google Maps, Street View, and Earth). For more information, see our copyright guidelines: http://journals.plos.org/plosone/s/licenses-and-copyright.

a. You may seek permission from the original copyright holder of Figures 1-3 to publish the content specifically under the CC BY 4.0 license.  

Reviewers' comments:

Reviewer's Responses to Questions

**Comments to the Author**

1. Is the manuscript technically sound, and do the data support the conclusions?

Reviewer #1: Partly

Reviewer #2: Yes

2. Has the statistical analysis been performed appropriately and rigorously? 

Reviewer #1: Yes

Reviewer #2: I Don't Know

3. Have the authors made all data underlying the findings in their manuscript fully available?

Reviewer #1: Yes

Reviewer #2: Yes

4. Is the manuscript presented in an intelligible fashion and written in standard English?

Reviewer #1: No

Reviewer #2: Yes

5. Review Comments to the Author

Reviewer #1: Dear editor I have interestingly read the manuscript and the followings are my suggestions for further improvement to be made by the authors.

In my opinion the study is not suitable for publication in the current form and suggest the followings to be deserved for publishing:

Abstract

I would like to add a few comments on the grammar of the abstract:

• In the first sentence, the phrase "right down to the present day" is a bit awkward. It would be better to say "to the present day" or "up to the present day".

• In the second sentence, the phrase "resolves much of the uncertainty" is a bit vague. It would be better to say something like "provides a clearer picture of" or "narrows down the range of uncertainty for".

• In the third sentence, the phrase "all primary sources with coverage that begins in March of 2003" is a bit wordy. It would be better to say "all primary sources that cover the period from March 2003 onwards".

Introduction

• The introduction is well-organized and provides a clear overview of the paper's purpose and scope.

• The author does a good job of summarizing the existing literature on the topic and highlighting the gaps in knowledge.

• The author's thesis statement is clear and concise.

• The introduction is grammatically correct and free of errors.

Methods and Materials

Here are some specific scientific questions that I have about the Methods and Materials section:

• How did the author account for the different sampling methods used in the different surveys?

• How did the author account for the different biases that may be present in the different surveys?

• How did the author validate the bootstrapping method for the type of data used in this study?

• How did the author choose the data sources for the study?

• How did the author analyze the data?

• What are the limitations of the study?

• How can the findings of the study be generalized to other populations?

Results

• Why did the author choose to use the 95% bootstrapped uncertainty intervals in the tables?

• What are the strengths and limitations of the bootstrapping method?

• How did the author account for the different sampling methods used in the different surveys?

• How did the author account for the different biases that may be present in the different surveys?

• How did the author validate the bootstrapping method for the type of data used in this study?

• How confident can we be in the estimates of violent deaths produced by the different surveys?

• What are the implications of the findings of this study for our understanding of the human costs of the Iraq War?

Discussion

1. Comparative Validation:

How does the rarity of the validation of three distinct methodologies (household-based survey estimation, sibling-based survey estimate, and casualty recording) impact the overall confidence in the study's findings?

Could you elaborate on the significance of the comparable validation found in [35] regarding violent deaths in the Kosovo war? How does this validation contribute to understanding the validity of the methodologies used in this study?

2. Quantifying Uncertainty:

How were the 95% uncertainty intervals calculated for individual surveys, and how do these intervals contribute to understanding the remaining uncertainty despite the consistency across group-1 sources?

Could you discuss the reasoning and methodology behind the back-of-the-envelope calculation used to estimate uncertainty for IBC-extended figures post-June 2011? How does this calculation help to address the absence of post-June-2011 surveys?

3. Validation of IBC Data:

What factors contribute to the consistent alignment of IBC-extended figures with mainstream sources? How does this consistency bolster the acceptance of IBC-extended numbers as a reliable guide to post-June-2011 violent death tolls?

Considering the change in IBC methodology in July 2011, how was the impact of this change evaluated, and how did it influence the reliability of the IBC-extended numbers?

4. Quantification and Credibility:

How does the tentative conclusion regarding the IBC-extended figures' accuracy, drawn from the ratios of IBC-extended figures to group-1 survey central estimates, enhance the credibility of the study's findings?

Could you elaborate on how the median and mean ratios of the back-of-the-envelope calculations contribute to the confidence in the accuracy of the IBC-extended figures?

5. Project Implications:

How does the validation of IBC data support the choice of the Costs of War Project to utilize IBC as a basis for quantification of the human cost of the Iraq War and dismiss the group-2 surveys?

In light of the disagreement between [36] and the current study's findings, how does the paper's validation contribute to maintaining credibility in quantification projects, and how can such projects avoid exaggeration?

6. Holistic Understanding:

How does the validation of IBC data contribute to the broader understanding of the Iraq War's human cost, as emphasized by [37] and [38], by providing a more accurate representation of individual victims beneath the abstract numbers?

Could you discuss how the validation of IBC data enhances the value of [18]'s casualty recording treatment, and how this approach provides a comprehensive perspective on the Iraq War's impact?

Reviewer #2: thank you very much for giving me the opportunity to review the manuscript: The Violent Death Toll from the Iraq War: 2003 - 2023. the manuscript has been written well and is easy to follow. I recommend it to be published.

6. PLOS authors have the option to publish the peer review history of their article (what does this mean?). If published, this will include your full peer review and any attached files.

Reviewer #1: **Yes: **Berun Anwar Abdalla

Reviewer #2: No

---

## [Author Response · Author response to Decision Letter 0]

8 Jan 2024

January 7, 2024

Dear editor,

I would like to thank you and the referee for a beautiful set of referee remarks. I took me a long time to respond to them, but the effort was worth it. The paper is much improved. Below I give point-by-point responses to the comments with indications of how they have affected the paper. 

Thank you very much for your consideration.

Sincerely,

Michael Spagat

Dear editor I have interestingly read the manuscript and the followings are my suggestions for further improvement to be made by the authors.

In my opinion the study is not suitable for publication in the current form and suggest the followings to be deserved for publishing:

Abstract 

I would like to add a few comments on the grammar of the abstract:

• In the first sentence, the phrase "right down to the present day" is a bit awkward. It would be better to say "to the present day" or "up to the present day".

• In the second sentence, the phrase "resolves much of the uncertainty" is a bit vague. It would be better to say something like "provides a clearer picture of" or "narrows down the range of uncertainty for".

• In the third sentence, the phrase "all primary sources with coverage that begins in March of 2003" is a bit wordy. It would be better to say "all primary sources that cover the period from March 2003 onwards".

[Response] Thank you very much. I have integrated these improvements into the text.

Introduction 

• The introduction is well-organized and provides a clear overview of the paper's purpose and scope.

• The author does a good job of summarizing the existing literature on the topic and highlighting the gaps in knowledge.

• The author's thesis statement is clear and concise.

• The introduction is grammatically correct and free of errors.

[Response] Thank you very much. I appreciate your kind words.

Methods and Materials

Here are some specific scientific questions that I have about the Methods and Materials section:

• How did the author account for the different sampling methods used in the different surveys?

I do not adjust author calculations based on sampling considerations. Rather, I critique sampling methods, when appropriate, so that readers can appreciate their weaknesses and likely biases and make their own mental adjustments.

For example, I criticize (Roberts et al. 2004) and (Burnham et al. 2006) for sampling primarily near main highways and streets, respectively, but do not adjust the numbers based on these criticisms. Instead, I present the unadjusted estimates alongside the critiques and then suggest that sampling bias as a possible reason why these estimates are so much higher than the other ones. 

For additional clarity I have added the following sentence to the text, referring to Roberts et al. 2004) I write:

“I judge this sampling bias to be severe but simply note the problem without attempting to adjust for it.”

Just above this sentence there is a reference to an article (Johnson et al. 2008) about sampling bias in (Burnham et al. 2006). I could add a similar sentence about how I didn’t try to adjust for this sampling bias either however I judge that such an addition to be unnecessary given that later I refer to much more serious evidence of fabricated data and ethical violations in (Burnham et al. 2006) that I note but do not adjust for. It strikes me as excessive to also toss in sampling bias here but I’m happy to do it if the referee prefers that I do so.

• How did the author account for the different biases that may be present in the different surveys?

The answer here is like the answer about sampling bias. In almost all cases I note issues but do not attempt to adjust figures in response to these issues. For example, I report that in the supposedly household-based ORB survey there are serious anomalies that reveal themselves when it is compared to other extended-family-based ORB surveys which show lower death rates for extended families than the survey in question does for households which are much narrower groupings than extended families are. In the paper I note the problem and then present the official estimates which have the anomaly built into them. There is, in fact, no viable alternative to reporting and criticizing the estimate since ORB has not released its raw data. This is what I now write in the paper:

“In short, ORB's Iraq work is not credible. Nevertheless, I still include it in the present paper, again to place it into perspective. I use ORB's published figures since ORB has not released the data that I would need to make an independent estimate.”

The Iraq Family Health Survey provides the only departure from the general rule of criticizing, but not correcting for, biases. I argue that the missing cluster adjustment and “arbitrary fudge”, which are built into the IFHS estimate, probably raise the estimate excessively. But for the IFHS I am able to provide two sets of estimates – one with and one without these adjustments. Figure 1 in the paper gives adjusted IFHS estimates and figure 2 gives unadjusted IFHS estimates.

• How did the author validate the bootstrapping method for the type of data used in this study?

I use bootstrapping to quantify the uncertainty surrounding survey-sample-based estimates of mean death rates. It is generally known, e.g., see (Rosenblum and Laan 2009) that this method is accurate as long as the distribution of sample observations is a reasonably good match for the population distribution. This condition can fail in small samples, especially when the underlying population distributions is highly skewed. Of the surveys covered in the paper, only Roberts et al. and Burnham et al. run afoul of these restrictions. Rosenblum and van der Laan (2009), therefore, propose their own method for constructing confidence intervals based on such data. I have decided, therefore, for the present version of the paper to augment figures 1 in the paper by including the Rosenblum-van der Laan confidence interval for the Burnham et al. paper. In addition, I have modified the text when bootstrapping is introduced to the following:

“Bootstrapping will yield accurate uncertainty intervals when the sample distribution mirrors the population distribution reasonably well, a condition that can fail in small samples, especially for highly skewed population distributions (@rosenblum2009). Therefore, we provide a separate, wider, uncertainty interval derived in @rosenblum2009 for the @burnham2006 survey that accounts for the skewed data collected in that small survey.”

• How did the author choose the data sources for the study?

This is clarified in the text with the phrase “all primary sources that cover the period from March of 2003 onwards.”

• How did the author analyze the data?

Here are the key points.

1. I present all figures as cumulative starting at the beginning of the war. This makes them comparable with each other.

2. I use IBC’s own numbers for civilians and then extend them to include combatants since all the other sources include combatants. The combatant figures are almost always taken from figures separately presented by IBC but I fill a few gaps from other sources.

3. The rest of the data comes from the surveys. 

a. For the surveys that have released their data I produce my own estimates using standard methods, calculating a per capita in-sample death rate and multiplying this rate by the population of Iraq at an appropriate moment in time. I use a single source for my population numbers to create as much consistency as possible across the estimates (Sometimes these population numbers differ a bit from the population numbers used in the original sources). Whenever possible I calculate my own 95% uncertainty intervals using bootstrapping although, as noted above, I also use the calculation of Rosenblum and Laan (2009) for an alternative uncertainty interval for the Burnham et al. (2004) survey.

b. When the surveys have not released their raw data I use the official estimates

4. I do quality evaluations of each source, considering their weaknesses, biases, etc..

5. I display the results in two graphs.

I believe that the above information is present and clear in the paper although I would be happy to add any clarifications that the referee thinks would help. The main innovation for the present version of the paper is the addition of the new confidence interval from Rosenblum and Laan (2009).

What are the limitations of the study?

The main limitations of the study are the quantity and quality of the data. More data, especially after 2011, would be welcome as we only have only Iraq Body Count after that. But more pre 2011 data would also be welcome. 

The main quality issue concerns the extreme unreliability of the (Roberts et al. 2004), (Burnham et al. 2006) and ORB estimates. Much effort in the paper is required to dispel the misinformation created by these works. However, quality issues are not confined to these surveys. For example, all field teams worked under time pressure as they were operating in time of war and the Iraq Family Health Survey even had to abandon some of its clusters due to security problems. A further issue is that several of the surveys have not made their data available.

The above two paragraphs are reproduced almost verbatim at the end of the Discussion section in the revised version of the paper.

• How can the findings of the study be generalized to other populations?

The short answer is that the paper is primarily about Iraq and the results do not extend to other populations. 

However, the paper does provide rare validation evidence for two methods commonly used to measure violent deaths in war. This validation is, however, complicated by the fact that three of the surveys evidently overestimated by wide margins. So the findings suggest, reasonably enough, that only high-quality surveys are valid for measuring violent war deaths.

These points should be clear in the present version of the paper.

Results 

• Why did the author choose to use the 95% bootstrapped uncertainty intervals in the tables?

This is a simple method that most people can understand that does not rely on normality assumptions. It is valid except for small, especially small and skewed samples which means that it is valid for all the surveys used in the paper except for the (Roberts et al. 2004) and (Burnham et al. 2006) surveys. For the latter survey I now offer an alternative confidence interval taken from (Rosenblum and Laan 2009). For the former survey the bootstrapped confidence interval is already so wide that it hardly seems worthwhile to present a still wider one. These points should be clear from the following additional text, already produced above:

“Bootstrapping will yield accurate uncertainty intervals when the sample distribution mirrors the population distribution reasonably well, a condition that can fail in small samples, especially for highly skewed population distributions (@rosenblum2009). Therefore, we provide a separate, wider, uncertainty interval derived in @rosenblum2009 for the @burnham2006 survey that accounts for the skewed data collected in that small survey.”

• What are the strengths and limitations of the bootstrapping method?

Bootstrapping does not require normality or, indeed, any parametric assumption. The limitation is that it assumes that the distribution of the sample is a good approximation of the distribution of the population. This assumption can cause problems in small samples, particularly those drawn from skewed populations. Again, these points should be clear from the current version of the text.

• How did the author account for the different sampling methods used in the different surveys?

I do not adjust original author calculations based on sampling considerations. I do criticize the sampling in Roberts et al. (2004) and Burnham et al. (2006) study but do not adjust the numbers based on these criticisms. Rather, I present the unadjusted estimates but then suggest that sampling bias as a possible reason why these estimates are so much higher than the other ones. 

• How did the author account for the different biases that may be present in the different surveys?

As with sampling bias, my general approach is to highlight potential biases, comment on their likely impact but not make explicit adjustments. The one exception is the missing cluster adjustment for the Iraq Family Health Survey for which I have two sets of estimates – one with the adjustment in and another with the adjustment out. See also my responses to bullet points in the Materials and Methods section of the present document.

• How did the author validate the bootstrapping method for the type of data used in this study?

I use the bootstrap to quantify the uncertainty surrounding survey-sample-based estimates of mean death rates. Bootstrapped confidence intervals are valid if the sample is representative of the population. This condition can fail in small samples, especially for skewed underlying data. Of the surveys covered in the paper, only Roberts et al. and Burnham et al. run afoul of these restrictions. Rosenblum and Laan (2009) propose their own method for constructing confidence intervals on such data. We include their confidence interval for the Burnham et al. paper in the revised version of our paper. Again, see my responses to the bullet points in the materials and methods section of the present document.

• How confident can we be in the estimates of violent deaths produced by the different surveys?

We should have no confidence whatsoever in Roberts et al. (2004), Burnham et al. (2006) and the ORB survey. We should have good confidence in the other surveys. These judgments are based on the quality assessments made in the paper and in the fact that the other surveys are greatly consistent with each other and with the data of Iraq Body Count. I believe that these points should be clear in the present version of the paper.

• What are the implications of the findings of this study for our understanding of the human costs of the Iraq War?

Here is the text in the present version of the paper that sums up my conclusions for the human costs of the first 20 years of the war:

“These numbers lead to the following tentative conclusions about the human cost of the first 20 years of the war; 1. The IBC-extended figures, roughly 200,000 violent deaths might be 20% higher than the true numbers of violent deaths: 2. The true numbers of violent deaths might, at a stretch, be twice the IBC-extended figures: 3. The mean of the above ratios is 1.1 while the median ratio is 0.91 so the preponderance of evidence points toward the IBC-extended figures being about right.”

Discussion 

1. Comparative Validation:

How does the rarity of the validation of three distinct methodologies (household-based survey estimation, sibling-based survey estimate, and casualty recording) impact the overall confidence in the study's findings?

We could, of course, be more confident if all the methodologies had been validated numerous times. I have added the following text to the paper:

“More such validations in the future would enhance confidence both in the results of the present paper and in the methods used for measuring violent war deaths.!

Could you elaborate on the significance of the comparable validation found in [35] regarding violent deaths in the Kosovo war? How does this validation contribute to understanding the validity of the methodologies used in this study?

There were three measurements of the number of war deaths in Kosovo that were made using three different methodologies but that reached highly similar and compatible conclusions, both in the overall number of people killed and in the time path of the killing. 

Two of the methods applied in Kosovo, a household survey and a casualty recording project, were also applied in Iraq. If these two methods worked in Kosovo that suggests that the can work in Iraq. I have added the following text to the paper:

“These methodologies included a household survey and a casualty recording project so this Kosovo validation enhancing confidence in the results of the present paper.”

2. Quantifying Uncertainty:

How were the 95% uncertainty intervals calculated for individual surveys, and how do these intervals contribute to understanding the remaining uncertainty despite the consistency across group-1 sources?

I use bootstrapping whenever I can, i.e., whenever the data is available. More specifically, I use the original data collected by as survey to generate many simulated samples. I provide all of the code for these simulations so anyone can see exactly what I’ve done.

Yes, there is consistency, but the fact of consistency does not remove all uncertainty. We certainly cannot know the exact number of people killed from the data at hand. We can just give a credible range.

These points should be clear in the present version of the paper.

Could you discuss the reasoning and methodology behind the back-of-the-envelope calculation used to estimate uncertainty for IBC-extended figures post-June 2011? How does this calculation help to address the absence of post-June-2011 surveys?

I clarify this point in the paper by adding the following text:

“During the pre-June-2011 period., i.e., when there are surveys we have 95% confidence intervals surrounding these surveys. The IBC-extended figures go right through the middle of the central estimates of these surveys so we can take the uncertainty intervals surrounding the survey central estimates as characterizing the uncertainty around the IBC figures up until June-2011. I then project this uncertainty forward.”

3. Validation of IBC Data:

What factors contribute to the consistent alignment of IBC-extended figures with mainstream sources? How does this consistency bolster the acceptance of IBC-extended numbers as a reliable guide to post-June-2011 violent death tolls?

IBC and the surveys are measuring the same thing, albeit using different methods. If they are all doing a reasonably good job of measuring violent deaths in the Iraq War, then they should be well aligned. This point should be clear in the present version of the text.

Considering the change in IBC methodology in July 2011, how was the impact of this change evaluated, and how did it influence the reliability of the IBC-extended numbers?

This is a misunderstanding. IBC did not change its methodology in 2011. I have clarified this point by tweaking the text that appears right before the Discussion section of the paper:

“Based on the consistent progression of IBC-extended through the middle of the group-1 surveys during the periods covered by the surveys and the fact that IBC did not change its methodology after 2011 I would suggest that IBC-extended remains a good guide to the violent death toll of the war after June of 2011.”

4. Quantification and Credibility:

How does the tentative conclusion regarding the IBC-extended figures' accuracy, drawn from the ratios of IBC-extended figures to group-1 survey central estimates, enhance the credibility of the study's findings?

It provides strong confidence in the figures up through June 2011 and decent confidence in the IBC after that period.

Could you elaborate on how the median and mean ratios of the back-of-the-envelope calculations contribute to the confidence in the accuracy of the IBC-extended figures?

These calculations are meant to show a range of reasonable figures pre-June-2011. We then assume that such a range continues to apply going forward. This point should be clear in the current version of the paper.

5. Project Implications:

How does the validation of IBC data support the choice of the Costs of War Project to utilize IBC as a basis for quantification of the human cost of the Iraq War and dismiss the group-2 surveys?

The Costs of War Project’s Iraq figures are taken straight from IBC so validation for IBC is also validation for Cost of War’s choice to use IBC. I believe that this point is clear in the text but I’m happy to add further clarification if the referee would like.

In light of the disagreement between [36] and the current study's findings, how does the paper's validation contribute to maintaining credibility in quantification projects, and how can such projects avoid exaggeration?

I clarify these points in the paper by adding the following text to the conclusions:

“Inevitably, some scientific papers are wrong and public trust in science requires that these errors are exposed and discarded. The existence of dueling survey-based estimates of violent deaths in the Iraq war has damaged the credibility of the field while the present paper should help to restore some this precious commodity. Transparency is a central ingredient to the scientific vetting process, and I have relied on the availability of some, alas not all, of the crucial data in writing the present paper. Hopefully, the field of war-death estimation will move toward greater transparency in the future so that more such work will be possible.”

6. Holistic Understanding:

How does the validation of IBC data contribute to the broader understanding of the Iraq War's human cost, as emphasized by [37] and [38], by providing a more accurate representation of individual victims beneath the abstract numbers?

I have tweaked the text in the final paragraph of the paper to clarify the answer to this question:

“They advocate telling the stories of the many individual victims who lie beneath the abstract numbers. @hamourtziadou2021 does precisely this, telling the stories of many of the individuals whose deaths are incorporated into the IBC database.”

Could you discuss how the validation of IBC data enhances the value of [18]'s casualty recording treatment, and how this approach provides a comprehensive perspective on the Iraq War's impact?

[18] is based on the IBC data and is written by someone who works to IBC. If, for example, the outcome of the validation exercise had been that IBC only coverts 1/10 of all the violent deaths in the war then, in effect, [18] would only be telling the stories of 1/10 of the victims. Maybe the other 9/10 have, qualitatively, very different stories. So knowing that, broadly speaking, IBC covers deaths as a whole in the conflict enhances the value of the stories in [18]. This is now the final sentence of the paper:

“The present paper's validation of IBC data can only enhance the value of this work because it shows that @hamourtziadou2021 provides, broadly, an overview of stories of all of the victims in the war.”

Bibliography for this Document

Burnham, Gilbert, Riyadh Lafta, Shannon Doocy, and Les Roberts. 2006. ‘Mortality after the 2003 Invasion of Iraq: A Cross-Sectional Cluster Sample Survey’. Lancet (London, England) 368 (9545): 1421–28. https://doi.org/10.1016/S0140-6736(06)69491-9.

Johnson, Neil F., Michael Spagat, Sean Gourley, Jukka-Pekka Onnela, and Gesine Reinert. 2008. ‘Bias in Epidemiological Studies of Conflict Mortality’. Journal of Peace Research 45 (5): 653–63.

Roberts, Les, Riyadh Lafta, Richard Garfield, Jamal Khudhairi, and Gilbert Burnham. 2004. ‘Mortality before and after the 2003 Invasion of Iraq: Cluster Sample Survey’. Lancet (London, England) 364 (9448): 1857–64. https://doi.org/10.1016/S0140-6736(04)17441-2.

Rosenblum, Michael A., and Mark J. van der Laan. 2009. ‘Confidence Intervals for the Population Mean Tailored to Small Sample Sizes, with Applications to Survey Sampling’. The International Journal of Biostatistics 5 (1): Article 4. https://doi.org/10.2202/1557-4679.1118.

---

## [Decision Letter · Decision Letter 1]

16 Jan 2024

The Violent Death Toll from the Iraq War: 2003 - 2023

PONE-D-23-23835R1

Dear Dr. Spagat,

We’re pleased to inform you that your manuscript has been judged scientifically suitable for publication and will be formally accepted for publication once it meets all outstanding technical requirements.

Kind regards,

Masoud Behzadifar

Academic Editor

PLOS ONE

Additional Editor Comments (optional):

Reviewers' comments:

Reviewer's Responses to Questions

**Comments to the Author**

1. If the authors have adequately addressed your comments raised in a previous round of review and you feel that this manuscript is now acceptable for publication, you may indicate that here to bypass the “Comments to the Author” section, enter your conflict of interest statement in the “Confidential to Editor” section, and submit your "Accept" recommendation.

Reviewer #1: (No Response)

2. Is the manuscript technically sound, and do the data support the conclusions?

Reviewer #1: (No Response)

3. Has the statistical analysis been performed appropriately and rigorously? 

Reviewer #1: (No Response)

4. Have the authors made all data underlying the findings in their manuscript fully available?

Reviewer #1: (No Response)

5. Is the manuscript presented in an intelligible fashion and written in standard English?

Reviewer #1: (No Response)

6. Review Comments to the Author

Reviewer #1: Thanks, dear author for your response, I am satisfied with your answers. The manuscript shows commendable clarity and depth in presenting the findings of his research.The author has effectively answered the questions presented, making a meaningful contribution to the field. The writing is crisp and concise, enhancing the overall readability of the manuscript. Acceptance is recommended, given strong alignment with scientific standards. The importance of the study and its potential impact on the field make it a valuable addition to the journal.

7. PLOS authors have the option to publish the peer review history of their article (what does this mean?). If published, this will include your full peer review and any attached files.

Reviewer #1: **Yes: **Berun Anwar Abdalla

---

## [Editor Report · Acceptance letter]

17 Feb 2024

PONE-D-23-23835R1 

PLOS ONE

Dear Dr. Spagat, 

I'm pleased to inform you that your manuscript has been deemed suitable for publication in PLOS ONE. Congratulations! Your manuscript is now being handed over to our production team.

Kind regards, 

on behalf of

Dr. Masoud Behzadifar 

Academic Editor

PLOS ONE